# The Recovery Benefit on Skin Blood Flow Using Vibrating Foam Rollers for Postexercise Muscle Fatigue in Runners

**DOI:** 10.3390/ijerph17239118

**Published:** 2020-12-06

**Authors:** Yi-Horng Lai, Ai-Yi Wang, Chia-Chi Yang, Lan-Yuen Guo

**Affiliations:** 1School of Mechanical and Electrical Engineering, Xiamen University Tan Kah Kee College, Zhangzhou 363105, China; lai81@xujc.com; 2Department of Sports Medicine, College of Medicine, Kaohsiung Medical University, Kaohsiung 807, Taiwan; aiyi0703@gmail.com; 3The Master Program of Long-Term Care in Aging, College of Nursing, Kaohsiung Medical University, Kaohsiung 807, Taiwan; chiachiyang@kmu.edu.tw; 4Program in Biomedical Engineering, College of Medicine, Kaohsiung Medical University, Kaohsiung 807, Taiwan; 5Department of Medical Research, Kaohsiung Medical University Hospital, Kaohsiung 807, Taiwan

**Keywords:** vibrating roller, recovery of muscle fatigue, skin blood flow, blood flow oscillation

## Abstract

Purpose: To determine the effect of vibrating rollers on skin blood flow after running for recovery from muscle fatigue. Method: 23 healthy runners, aged between 20 to 45 years, participated in a crossover trial. Muscle fatigue was induced by running, and recovery using a vibrating roller was determined before and after the intervention. Each subject was measured at three time points (prerun, postrun, and postroller) to compare skin blood flow perfusion and blood flow oscillation at the midpoint of the dominant gastrocnemius muscle. The results show that blood perfusion is greater when a vibrating roller is used than a foam roller, but there is no statistical difference. The analysis of blood flow oscillation shows that vibrating rollers induce 30% greater endothelial activation than a foam roller. Vibrating rollers significantly stimulate the characteristic frequency for myogenic activation (*p* < 0.05); however, the effect size is conservative.

## 1. Introduction

For prolonged exercise, leg muscles undergo repetitive contraction patterns over a long time. During long-term exercise, the main mechanism for muscle damage is the constant contraction and relaxation of muscle fibers, which damages muscle filaments, destroys the sarcoplasmic reticulum, increases lactic acid accumulation, and decreases ATP energy. This type of exercise pattern often leads to physical and mental fatigue [1,2].

Nervous system fatigue is defined in terms of central nervous system fatigue or peripheral nerve fatigue. The main mechanism involves interaction between the nervous system and muscle cells. Psychological stress and endocrine factors affect motor nerve fibers, muscle fibers, and muscle sensory receptors. The reaction in the motor cortex decreases the excitement of nerve impulses that are transmitted to the spine, and there are fewer potassium ions in nerve cells [3].

Manual massage after exercise can reduce muscle fatigue [4,5,6]. Physical stimulation of muscles by massage increases local blood circulation and removes metabolic waste. As manual massage is limited in terms of the execution angle for massage and efficiency, massage tools (e.g., massage balls, yoga columns, and foam rollers) have been developed. A foam roller (FR) can be used for self-myofascial release, which is relatively convenient, effective, and affordable [7,8,9,10,11]. On the other hand, in [12], the effects of massage were rather small and only relevant in the short term. The performance recovery of massage remains partly unclear.

Self myofascial release uses “autogenic inhibition” [13]. During the rolling massage period, the pressure that is applied to the muscles stimulates the sensory receptors (i.e., Golgi’s body) at the junction of the muscles and the muscle bonds. Golgi’s body detects the variation in muscle tension, and the muscle spindle regulates the length of muscle fibers to relax the muscles. The protective mechanism of autogenic inhibition prevents injuries to the muscles. The experiment results for self myofascial release show that the VAS pain scale scores, muscle tension, and stiffness are reduced, and the range of joint motion, muscle elasticity, and softness of fascia is increased [14].

Vibration therapy stimulates the H reflex of muscles, recruits more motor units, and activates nerve receptors. Studies have confirmed that vibration stimulation reduces postexercise fatigue and muscle soreness [15,16]. In addition, the acute physiological reaction mechanism for vibration increases blood circulation in the local muscles [17]. Vibration stimulation may also improve jumping performance and agility [18].

Combining the benefits of foam rollers and vibration stimulation, vibrating rollers (VRs) are used for self-myofascia relaxation. Vibration stimulation stimulates Golgi’s keys, inhibits muscle contraction, and promotes muscle relaxation. Studies involving VRs show that VRs also increase the flexibility of muscles and the angle of joint movement and reduce pain [19,20,21,22,23,24,25,26,27].

For microcirculatory, skin blood flow (SBF) provides tissues with vital oxygen and nutrients while removing waste products and distributing signaling molecules around the organism. Previous studies have shown that vibration stimulation increases muscle temperature and skin blood. In addition, SBF reduces delayed muscle soreness after exercise. This acute physiological reaction mechanism is involved in muscle recovery [28,29,30,31,32,33]. Furthermore, blood flow oscillation (BFO) reflects the current functional state of blood flow regulation systems. BFO analysis shows that characteristic frequencies reflect endothelial activity (0.008–0.02 Hz), sympathetic neurogenic activity (0.02–0.06 Hz), myogenic activity (0.06–0.2 Hz), respiratory activity (0.2–0.6 Hz), and cardiac activity (0.6–2.3 Hz) [34,35,36,37,38,39].

At present, most studies involving FRs focus on deep fascia massage. Whether the use of VRs to stimulate local soft tissue can promote micro blood flow and accelerate muscle fatigue recovery is still unknown. In this study, we hypothesize that vibration stimulation can stimulate Golgi’s bond, inhibit muscle contraction, and promote muscle relaxation. Therefore, VRs may be more able to relax the muscle fascia than FRs. In addition, VRs may have lower pain intensity than FRs when performing self fascia relaxation. This study uses laser Doppler flowmetry measurements to determine the effects of VRs on local SBF and compares the effects of FRs and VRs on BFO. The effectiveness of muscle recovery after fatigue due to exercise is also determined.

## 2. Materials and Methods

All experimental procedures were approved by the Institutional Review Board of Kaohsiung Medical University Chung-Ho Memorial Hospital (No. KMUHIRB-E(I)-20190022). The protocol for experiments complies with the relevant guidelines of the Declaration of Helsinki. Written informed consent was obtained from each voluntary participant.

### 2.1. Participants

A total of 23 healthy novice runners (11 women and 12 men) were recruited for this study. The criteria for subjects included being aged between 20 and 45 years old and running regularly from 1 to 3 times a week at a speed of at least 6 km/h for 5–10 km and/or 30–50 min each time. Exclusion criteria included pulmonary disease, acute infection, cardiovascular disease, musculoskeletal injuries within 6 months, leg fracture surgery, diseases with neurological symptoms, varicose veins, and lower limb pain. Table 1 summarizes the demographic data for participants.

### 2.2. Examination Protocol

This study uses a crossover experiment. The flow diagram of the protocol is shown in Figure 1. The experimental group and the control group are the same group of subjects. The order of experimental intervention was decided randomly, so there are no individual differences between the two groups. The protocol includes a baseline, fatiguing exercise and recovery with intervention. Skin blood perfusion was measured before (prerun) and after (postrun) a 50-min fatigue-inducing run [2], whereby the participant begans to walk briskly on a treadmill (without a gradient) at a speed of 6 km/h. After warming up for 5 min, the subject accelerated at a rate of 1 km/h every 2 min until Borg’s score reached 13 (somewhat hard). Then, the subject ran at a steady speed for 50 min until Borg’s score reached 17 (very hard) or 90% of the maximum heart rate (i.e., HRmax = 200 − age). Lastly, the subject continued to run for another 2 min, before slowing and finishing the run.

After completing postrun measurements, participants performed 6 min of VR or FR, which was randomized for recovery intervention and SBF measurement (postroller). The participants had a one-week interval to recover from muscle fatigue between crossover experiments.

For roller recovery treatment, participants were supported on the ground with their arms and pushed their bodies forward and backward from the knee fossa to the Achilles tendon for 3 s and then reversed from the Achilles tendon to the knee fossa. The entire calf gastrocnemius muscle and the medial and lateral muscles of the gastrocnemius muscles of both legs were massaged. Each leg was massaged for 3 min (Figure 2). The visual analog scale (VAS) was used to evaluate the pain intensity of subjects during the intervention.

### 2.3. Examination Instrument and Outcome

#### 2.3.1. Vibrating Roller Technique

The VR (FE07-VIBR50), designed by LAIN HONG SHING YEH CO, Taiwan, has a 5-stage preset vibration frequency (20, 25, 32, 40, and mixed frequency). The vibration frequency selected in this study is the mixed frequency mode (i.e., 20~40 Hz: 20 Hz → 25 Hz → 32 Hz → 40 Hz → 32 Hz → 25 Hz → 20 Hz → 25 Hz; automatic adjustment every 10 s) (Figure 3).

#### 2.3.2. Laser Doppler Flowmetry

SBF perfusion was measured using a laser Doppler flowmeter (Oxford Optronix Ltd., Abingdon, UK). The sampling frequency is 256 Hz. The measured signals are in relative units (blood perfusion units (BPU)). Each subject was subject to a 2-min SBF measurement before running, after running, and after using a VR or FR. The subjects were supine and kept a fixed posture during the measurements to reduce artificial interference. The measurement site was estimated visually at the midpoint of the calf gastrocnemius muscle of the dominant leg (Figure 4).

#### 2.3.3. Blood Flow Perfusion and Oscillation Analysis

Using Matlab 2016 (MathWorks Inc., Natick, MA, USA), the signal noise was filtered and the SBF perfusion was calculated. Similar to previous studies [34,35,36,37,38], a Morlet wavelet transform was used to determine BFO. Five characteristic frequencies in the frequency interval between 0.0095 and 2 Hz were identified. The outcomes of BFO analysis reflect the activity of metabolic endothelial-related controls (0.008–0.02 Hz), sympathetic neurogenic systems (0.02–0.06 Hz), myogenic activity (0.06–0.2 Hz), respiratory movements (0.2–0.6 Hz), and cardiac movements (0.6–2.3 Hz).

Examples of SBF perfusion and BFO are shown in Figure 5. The frequency corresponding to the scale of the Morlet wavelet parameter ranged from 0.007 to 2.62 Hz over 50 intervals. The respective scale boundaries for the characteristic frequency were at 306, 1033, 3506, and 10530. Vibrating intervention affects only a local area of skin and causes no local heating. We anticipate that the interventions are local mechanisms of reactive hyperemia. Only three characteristic frequencies (endothelial, neurogenic, and myogenic) were analyzed.

### 2.4. Statistical Analysis

Data are expressed as mean and standard deviation. All results were analyzed using SPSS 25 (SPSS Inc., Chicago, IL, USA) statistical software. A two-way repeated-measures ANOVA and a posthoc analysis were conducted to determine the effects of different conditions on dependent outcomes. Any significant interaction × time was identified and a posthoc analysis was used to determine the effect.

Furthermore, paired *t*-tests within each group were conducted to determine the significant effects of the intervention. In addition, the relative changes in normalized measures from prerun to postrun and from postrun to postroller were analyzed using a paired *t*-test. The significance difference (α) was set at 0.05.

## 3. Results

Using MATLAB and data analysis, the data for SBF perfusion and BFO at three time points (i.e., prerun, postrun, and postroller) are calculated.

### 3.1. Skin Blood Flow Perfusion

#### 3.1.1. Two-Way Repeated-Measures ANOVA and Posthoc Analysis in Skin Blood Flow Perfusion

The results for SBF perfusion prerun, postrun, and postroller are shown in Table 2 and Figure 6. The ANOVA result is shown in Table 3. The posthoc analysis results are shown in Table 4.

After using VR, SBF perfusion was higher for the VR group than the FR group. There were larger individual differences in the SBF perfusion, so the difference between the two types of rollers was not significant. The ANOVA analysis shows a significant time effect (*p* < 0.001). The posthoc test results show a significant increase between prerun and postrun measurements for the FR group (Table 4).

#### 3.1.2. Analysis of Relative Changes in Normalized Skin Blood Flow Perfusion

The paired *t*-tests for relative changes in normalized SBF perfusion for the two types of rollers are shown in Table 5. For the VR group, there is a significant effect of relative change in normalized perfusion between prerun and postroller measurements. However, the relative change in normalized perfusion between postrun and postroller measurements is not significant. For the FR group, the results in Table 4 and Table 5 show a significant improvement between prerun and postrun measurements. As the intervention time increases, the relative change in normalized perfusion gradually decreases.

### 3.2. Blood Flow Oscillation

#### 3.2.1. Two-Way Repeated-Measures ANOVA and Posthoc Analysis in Blood Flow Oscillation

The spectrum energy of BFO for prerun, postrun, and postroller is shown in Table 2. The trend plot of normalized energy in myogenic frequency band is shown in Figure 7. The ANOVA analysis is shown in Table 3. In terms of endothelial and neurogenic energy, the ANOVA results are not significant for either the FR group or the VR group. However, for myogenic energy, the time effect has a significant effect.

#### 3.2.2. The Analysis of Relative Changes in Normalized Blood Flow Oscillation

The results for a paired *t*-test for the relative change in normalized BFO for the two types of rollers are shown in Table 5.

For normalized endothelial and neurogenic energy, the paired *t*-test results are not significant for either the FR group or the VR group. However, between postrun and postroller sections, the VR group shows a greater relative change (33%) in normalized energy than the FR group.

In terms of normalized myogenic energy, there is a significant effect of relative change between prerun and postroller measurements for the VR group.

### 3.3. Visual Analog Scale (VAS)

The result of VAS for FRs was 6.2 (2.11); for VRs, it was 5.72 (2.49). The pair *t*-test result was not significant. In addition, taking the number of people as the unit, during the intervention, 11 (48%) subjects felt less pain when using VRs, 7 (30%) subjects felt less pain when using FRs, and 5 (22%) felt the same pain intensity.

## 4. Discussion

The results for skin blood flow perfusion (Table 2) show that SBF perfusion for the FR group decreases for the postroller measurement. There is decreased perfusion for the FR group because when FR involves insufficient compression, the effect of self-myofascia relaxation is not consistent. Excessive muscle tension inhibits blood supply and slows muscle recovery [7]. For this study, a larger number of subjects reported that when using VRs for self-myofascia relaxation massage, the pain intensity during muscle pressing was less than that of FRs. This result is different from that of a previous study, in which FR and VR groups exhibited a higher pain threshold than the control group [12,20]. Some studies have noted that VRs or FRs can increase the muscle pain threshold, but only VRs achieve statistically significant improvement benefits [26], which is also demonstrated by this study.

For the VR group, blood flow perfusion increases, but not significantly. SBF that is induced by vibration has been reported by many previous studies [17,28,29,31,32,40], but the parameters for vibration (frequency, device mode, intervention time) are not constant. One previous study showed that after 5 min intervention with 30 Hz vibration, SBF gradually decreased. Different frequencies of vibration have a specific effect on SBF during the recovery period. A frequency of 50 Hz increases SBF, and a frequency of 30 Hz decreases SBF [17]. This study uses a 6-min protocol of VR intervention to determine the adaptability of participants, whereby the frequency of VR automatically changes every ten seconds (20 Hz → 25 Hz → 32 Hz → 40 Hz → 32 Hz → 25 Hz → 20 Hz → 25 Hz). This may explain why a local vibration device has no significant effect on SBF.

In one study [40], passive intervention using vibration was shown to increase SBF, but there was no increase in blood flow in the femoral artery or in skeletal muscle. The previous findings cannot be applied to explain the hemodynamics of muscle fatigue recovery. In this condition, blood flows away from the integumentary system and is redirected to the musculature. Another vibration study concluded that nitric oxide (NO) synthase in endothelial cells increases SBF. However, NO is only one of the endogenous factors that induce skin vasodilation. Epinephrine, norepinephrine, and histamine are also activated by vibration to promote a vasodilatory response [41]. For muscle fatigue exercise, the dominant regulation mechanism for SBF changes from sympathetic activation to hypothalamus regulation. Skin vasodilation opens all capillaries in skeletal muscles, and skeletal muscle blood flow increases to 80% of cardiac output [30]. The complicated relationship between endothelial, neurogenic, and myogenic activity on skin vasodilation can be determined using BFO analysis.

The wavelet analysis results for BFO show no significant difference between postrun and postroller results. However, the relative energy for endothelial activation increases by nearly 33% for the VR group. During muscle exercise, the balance between oxygen supply, oxygen demand, and the control of blood flow is determined by metabolic factors [33]. The results of this study are in agreement with those of past studies [34,41] in that the characteristic frequency of endothelial activation is 0.008 to 0.02 Hz and promotes SBF perfusion.

Myogenic activity is associated with the rhythmic oscillations of capillaries, so the decrease in myogenic activity in wavelet power reflects a decrease in SBF [35]. This study determines that VRs significantly stimulate the characteristic frequency for myogenic activation. The myogenic response, which is most common in smaller resistance arteries, refers to the contraction that is caused by the muscle cell itself. It is used to regulate organ blood flow and peripheral resistance. When the blood vessels are in the precontraction state, myogenic activation can cause additional contraction or expansion, which increases or decreases blood flow [42].

All participants for this study were healthy novice runners, and the procedure of fatigue-inducing running is exhausting, so there were few participants. There were no comparisons with a control group (a group without roller intervention) for this study. Second, pulse rate was measured during the measurement of SBF, but the pulse rate and respiration of subjects during the 50-min fatigue running were not, so the characteristic frequencies for respiratory movements (0.2–0.6 Hz) and cardiac movements (0.6–2.3 Hz) in BFO cannot be verified. Moreover, because of the conservative effect size, the result of the current work is exploratory and not confirmative. For these reasons, if the reader wants to translate the result into practice, the present findings should be interpreted with caution.

## 5. Conclusions

In this study, there was no significant difference in the SBF perfusion trend between FR and VR intervention for muscle fatigue recovery. Using a variable frequency VR, the relative change in normalized perfusion was not significant. There was greater myogenic activation in BFO after VR intervention. However, due to the conservative effect size, further research is needed to elucidate whether the variable frequency VR for self-fascia relaxation massage can accelerate recovery from fatigue.

## Figures and Tables

**Figure 1 ijerph-17-09118-f001:**
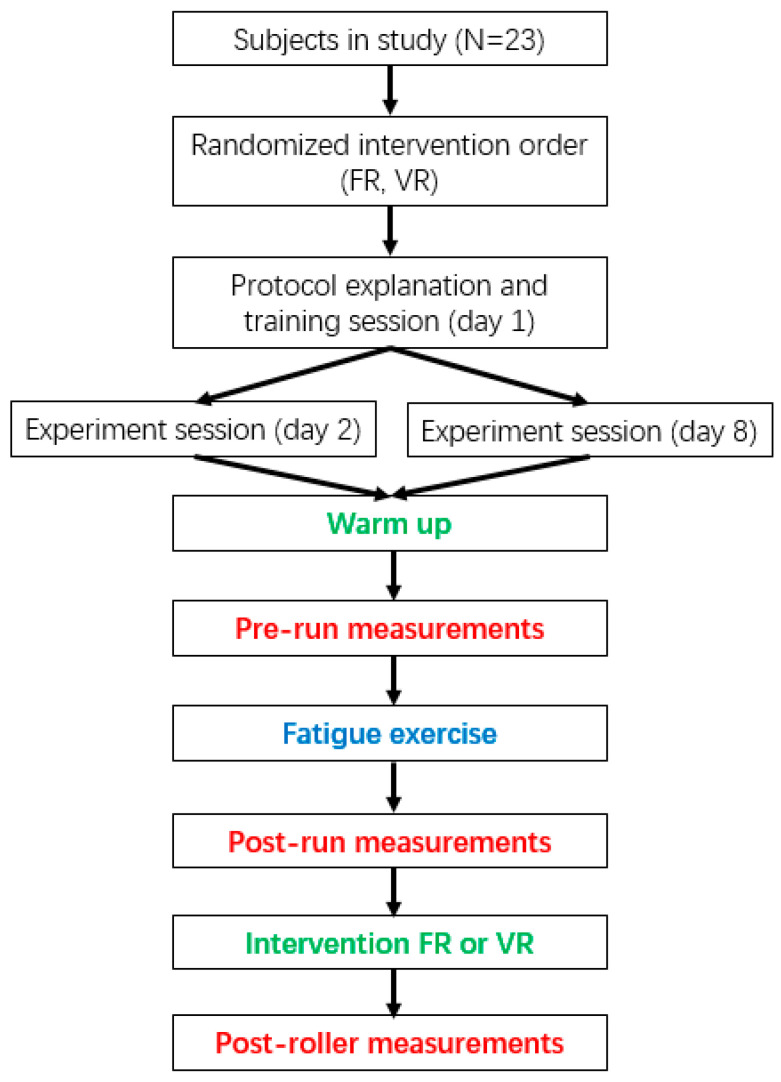
The flow diagram of the protocol.

**Figure 2 ijerph-17-09118-f002:**
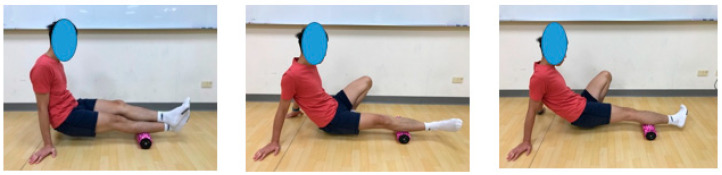
The procedure of roller treatment.

**Figure 3 ijerph-17-09118-f003:**
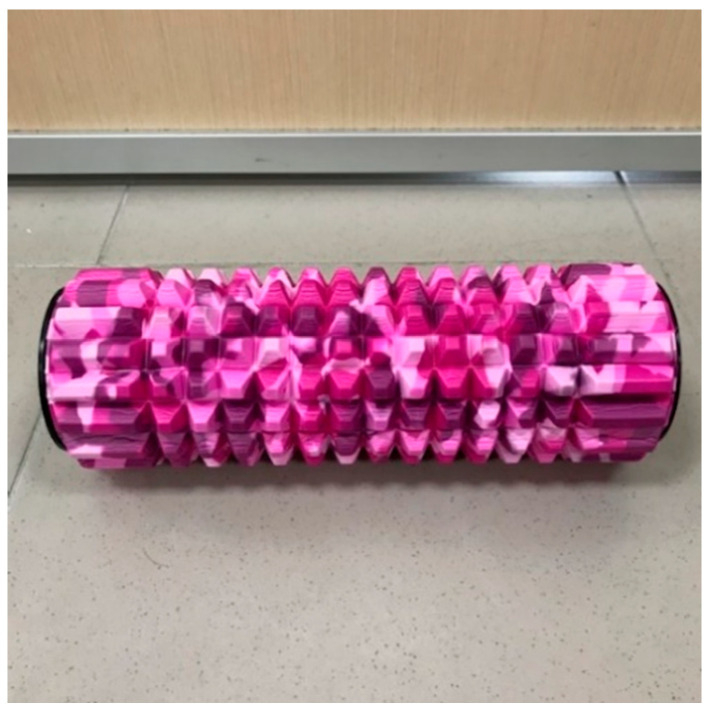
The vibrating roller.

**Figure 4 ijerph-17-09118-f004:**
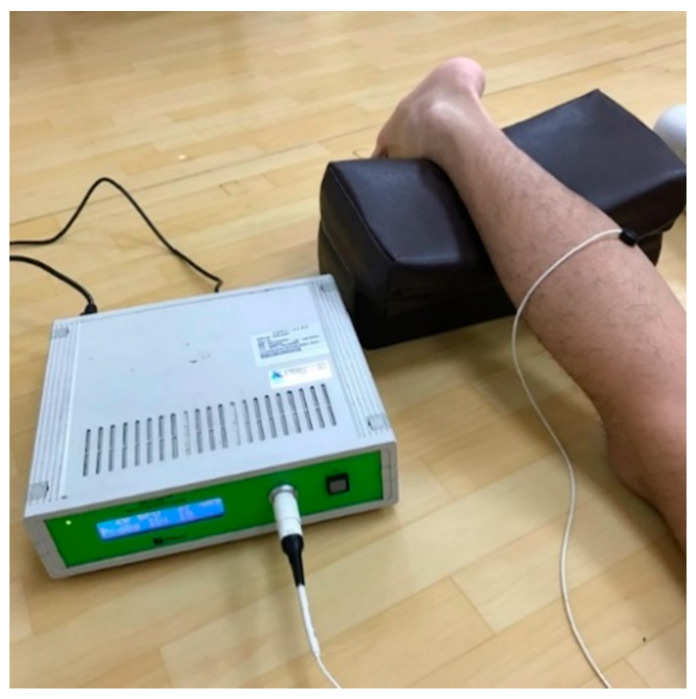
The skin blood flow measurement.

**Figure 5 ijerph-17-09118-f005:**
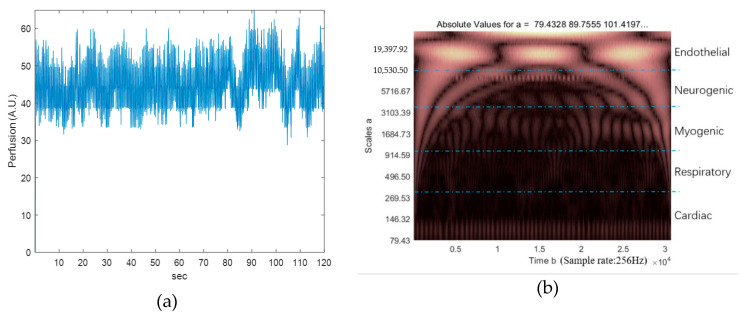
An example of typical records of skin blood flow perfusion (**a**) and blood flow oscillation spectrum (**b**).

**Figure 6 ijerph-17-09118-f006:**
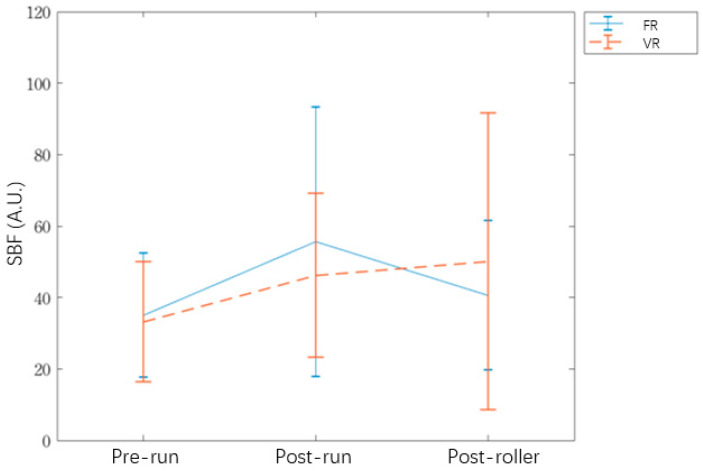
The SBF trend between prerun, postrun, and post-roller.

**Figure 7 ijerph-17-09118-f007:**
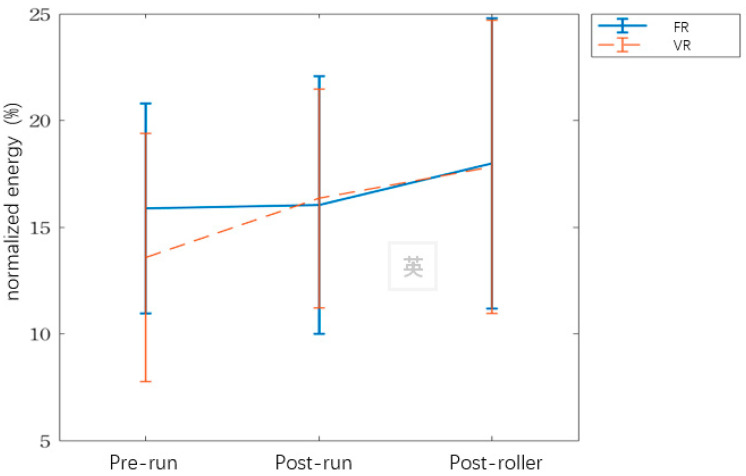
The trend plot of normalized energy in myogenic frequency band.

**Table 1 ijerph-17-09118-t001:** Characteristics of the demographic data.

Title 1	Male	Female
Age (years)	26.2 ± 5.2	26.6 ± 7.8
Height (m)	169.5 ± 3.9	156.8 ± 5.1

**Table 2 ijerph-17-09118-t002:** The experimental result of skin blood flow (SBF; A.U.) and normalized energy of blood flow oscillation (BFO; %).

		Prerun	Postrun	Postroller
SBF	Foam roller	35.1 (17.4)	55.6 (37.8)	40.7 (20.8)
	Vibrating roller	33.3 (16.8)	46.2 (23.0)	50.1 (41.6)
BFO endothelial	Foam roller	11.1 (6.0)	11.0 (5.3)	11.4 (5.4)
	Vibrating roller	10.5 (6.1)	10.6 (4.2)	12.5 (7.9)
BFO neurogenic	Foam roller	7.6 (3.0)	6.5 (4.1)	7.0 (4.0)
	Vibrating roller	7.0 (4.3)	5.6 (2.4)	6.3 (2.9)
BFO myogenic	Foam roller	15.9 (5.0)	16.0 (6.0)	18.0 (6.8)
	Vibrating roller	13.6 (5.8)	16.4 (5.1)	17.8 (6.9)

Values are mean(standard deviation).

**Table 3 ijerph-17-09118-t003:** Two-way ANOVA analysis of SBF and normalized energy of BFO.

		F Value	df	*p*	$\eta^{2}$
SBF	Roller	0.018	1	0.894	0.0001
	Time	9.810	2	0.001 *	0.1294
	Roller × Time	1.128	2	0.343	0.0168
BFO endothelial	Roller	0.163	1	0.690	0.0012
	Time	0.891	2	0.425	0.0133
	Roller × Time	0.283	2	0.756	0.0043
BFO neurogenic	Roller	2.931	1	0.101	0.0217
	Time	1.338	2	0.284	0.0199
	Roller × Time	0.031	2	0.970	0.0005
BFO myogenic	Roller	0.733	1	0.401	0.0055
	Time	4.439	2	0.025 *	0.0630
	Roller × Time	1.298	2	0.294	0.0193

* Significant difference (*p* < 0.05).

**Table 4 ijerph-17-09118-t004:** Two-way ANOVA posthoc analysis of SBF (A.U.).

Roller	Section (I)	Section (J)	Difference (J-I)	*p*
Foam roller	Prerun	Postrun	20.536 (7.130)	0.026 *
	Prerun	Postroller	5.579 (3.762)	0.457
	Postrun	Postroller	−14.957 (7.981)	0.223
Vibrating roller	Prerun	Postrun	12.988 (5.306)	0.068
	Prerun	Postroller	16.903 (7.890)	0.130
	Postrun	Postroller	3.915 (8.680)	1

* Significant difference (*p* < 0.05).

**Table 5 ijerph-17-09118-t005:** The analysis of relative changes in normalized SBF and BFO (%).

		Intervention	Pair Difference	df	*p*
SBF	Prerun vs.	Foam roller	27.90% (78.44)	22	0.04 *
	Postroller	Vibrating roller	79.23%(137.33)		
	Postrun vs.	Foam roller	−2.21% (97.52)	22	0.24
	Postroller	Vibrating roller	19.00%(91.59)		
BFO endothelial	Prerun vs.	Foam roller	33.10% (79.17)	22	0.32
	Postroller	Vibrating roller	46.47% (102.10)		
	Postrun vs.	Foam roller	19.38% (60.35)	22	0.12
	Postroller	Vibrating roller	53.08% (138.63)		
BFO neurogenic	Prerun vs.	Foam roller	−4.11% (58.59)	22	0.42
	Postroller	Vibrating roller	−7.16% (50.73)		
	Postrun vs.	Foam roller	12.77% (55.25)	22	0.27
	Postroller	Vibrating roller	23.61% (67.43)		
BFO myogenic	Prerun vs.	Foam roller	13.52% (34.34)	22	0.03 *
	Postroller	Vibrating roller	55.19% (99.48)		
	Postrun vs.	Foam roller	21.16% (72.93)	22	0.43
	Postroller	Vibrating roller	17.65% (45.58)		

* Significant difference (*p* < 0.05).

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
