# Peer review of "The Recovery Benefit on Skin Blood Flow Using Vibrating Foam Rollers for Postexercise Muscle Fatigue in Runners"

_ijerph, 2020, doi:10.3390/ijerph17239118_

Round 1

Reviewer 1 Report

The paper "The Recovery Benefit on Skin Blood Flow Using Vibrating Foam Rollers for The Post-Exercise Muscle Fatigue in Runners" is very interesting and its great novelty is undoubted.

Then, among POINTs of STRENGHTS we can list:

  • Novelty
  • Methods and in general statistics

On the contrary, before considering acceptance I'd like to suggest some improvements:

  • First of all, authors may read also the following article for ameliorating and enlarging current discussion
    • PeerJ. 2019 Nov 26;7:e8000. doi: 10.7717/peerj.8000. eCollection 2019.
  • Moreover, sample size is quite short and that makes final findings uncertain for translating into the practice.
  • Added to previous point, I think authors should "smooth" their conclusion identifying more clearly LIMITATIONS of the study.
  • Finally, references seems to be not in accordance with journal's rules. PLEASE CHECK!

Reviewer 2 Report

The current article investigates the influence of two different foam rollers on changes in SBF and BFO, more explicitly endothelial, neurogenic, and myogenic factors, after a highly fatigueing 60 min run. 

This is of interest to both the scientific community and the wider community interested in different recovery methods.

The methods and results section are relatively well written and I only have some minor comments. 

  1. Please include effect sizes and confidence intervals where relevant
  2. A large variance in individual responses is mentioned, please include a figure to illustrate this.
  3. I suggest moving some of the tables to the appendix and only including the most important results in the results section (along with the  descriptive stat already included in table 1)

Why did you choose an ANOVA, in my opinion, the most important data is to compare the post-run post-roller change between FR and VR. 

  1. Check you have units of measurement in all tables
  2. I suggest deleting line 146-147
  3. line 142, what three time point
  4. Manufacturer and country reference is missing for SPSS and Matlab
  5. Line 119, What filter was used and how the signals were transformed to SBF
  6. Line 115 specify how you found the mid point1
  7. line 102 lower leg or foot? 
  8. line 94, I don't quite understand this part of the protocol. 

In my opinion, both the introduction and discussion need major changes that better reflect the uniqueness and novelty of the current study

Introduction

  1. Please reduce all general text on fatigue and then increase the text starting 61-66. Include a thorough introduction to current research in this area (and if there are other measurement methods)
  2. In particular line 65-66 where I would like o know more about how changes in these variables are associated with improved recovery. That explains the hemodynamics of muscle fatigue recovery.
  3. Please, in a more clear way explain the difference between vibration therapy and vibration foam roller therapy. This will give a better idea of why the VFR is a novel and interesting tool.
  4. Much of the discussion could be included in the introduction. This will allow for a more comprehensive discussion of your results. Rather than the descriptive nature of what each variable measures that currently characterizes your current discussion

Discussion

  1. Early in the discussion you mention perceived pain, BUT you did not measure this (or at least report it) please delete this.
  2. What is endothelial activation energy? This is new for me
  3. Please include a section on limitations in the current study and recommendations for future research.

Reviewer 3 Report

This manuscript compared the skin blood flow and blood flow oscillation responses to two different post-exercise recovery aids, a foam roller and a vibrating roller, in a healthy novice runner population.  Following a fatiguing bout of running exercise, participants performed 6 minutes of a randomly assigned recovery aid in a cross-over fashion, followed by a post-roller measurement of skin blood flow and blood flow oscillation.  The results demonstrated a non-statistically significant trend towards an increase in blood perfusion with the vibrating roller and a 30% greater endothelial activation with the vibrating roller compared to the foam roller.  The authors concluded that “vibrating rollers significantly stimulates the characteristic frequency for myogenic activation and increases acute SBF perfusion.”  In present form, the manuscript suffers from incomplete and in some cases, unclear reporting, the statistical analysis is difficult to interpret and may need correction, and the reliance on a single outcome measure (laser Doppler flowmetry) limits the impact, reliability, and importance of the findings.  The conclusion appears to be biased towards the vibrating roller, in spite of conflicting data in the results.  The absence of a naïve control group (no intervention) is another limiting factor, which is briefly acknowledged but not discussed (Lines 231-232), and no justification of sample size is given despite acknowledgement of the small sample size for this study (Line 231).

Major Comments:

  • The methods and statistical reporting is incomplete and more detail is required throughout. How was the sample size determined (a priori power analysis?)?  The exercise protocol is not well described – presently in lines 91-94 the authors state that “the subject accelerated at a rate of 1 km/h every 2 minutes and ran at a steady speed for 50 minutes until Borg’s score reached 17 (very hard) or 90% of the maximum heart rate and running continued for another 2 minutes, before slowing and finishing the run”; however no detail is given how the steady speed was determined (fixed speed? % of predicted max HR? % of VO2max?), and it is unclear what the criteria for terminating exercise was (50 min? Borg RPE scale score of 17? 90% of max HR? whenever one of those was reached?)  Similarly, statistical reporting appears incomplete as well – with such a small sample size, were assumption for ANOVA tested and met?  Why were paired t-tests used for post-hot testing?  This could be a source of substantial error given that multiple comparisons were being made and repeated t-tests are not adjusted for multiple comparisons (unlike other post-hoc tests including Tukey’s, Bonferroni, etc.).  Similarly, in Lines 190-192 the authors state that “a larger number of subjects reported that when using a VR for self-fascial relaxation massage, the pain during muscle pressing was less than that for a FR”; however 1) no statistical analysis is given (e.g., Chi-squared or other categorical variable test statistic); and 2) no description of the scale used to assess pain during muscle pressing is given in the methods. If this is anecdotal, then it should be omitted given it was not collected and analyzed in a systematic and rigorous method, and if not, then the complete method should be described
  • A major limitation to this study is its reliance on a single outcome measure (laser Doppler flowmetry), with no functional outcome measure either. This is particularly concerning because a number of other techniques and validated scales are widely available to test post-exercise muscle fatigue, pain, and delayed onset muscle soreness.  For example, pressure pain threshold at standardized sites with an algometer can quantify muscle tenderness, rating scales are available for soreness, range of motion, swelling, and isometric torque/strength measures are commonly employed, as are more objective methods including potentiated quadriceps twitch force.  Please provide an explanation why no other outcome measures were employed
  • Conclusions do not appear to be supported by the results of this study

Minor Comments:

  • English language editing is required throughout.
  • Consider reviewing terminology throughout and revising for precision of language. For example, the authors repeatedly refer to “muscle fatigue recovery”, yet no measures of fatigue were recorded and reported; perhaps the authors are referring to the use of FR/VR as a post-exercise recovery aid?
  • Lines 30-34 – not sure why this is relevant; moreover, marathon running is not necessarily high-intensity, especially for less experienced and less-trained runners; prolonged exercise and high-intensity exercise are distinct and should not be conflated
  • Line 31 – “muscle fatigue” here is used to describe a multitude of physiologic processes that may not necessarily reflect the classical definition of fatigue (a transient reduction in force-generating capacity of the muscle reversible with rest) – for example, muscle damage is a distinct process; suggest rewording
  • Lines 35-39 – again, unsure why this description is here, as it does not appear to be salient to the present study
  • Lines 40-44 – while the cited work is supportive of manual massage as a recovery aid; suggest presenting a balanced overview to include other studies which have not shown a benefit of manual massage/foam roller/SMR on recovery (for example, Poppendieck W, et al. 2016. Massage and Performance Recovery: A Meta-Analytical Review. Sports Med 46:183-204)
  • Line 48-49 – how is this mechanism protective for injury? Also, please provide a citation to support this statement
  • Lines 52-56 – more clarity on how vibration therapy is distinct from and/or complementary to SMR would be helpful.
  • Line 52-53 – how are these mechanisms helpful in aiding recovery?
  • Lines 57-60 – how are these mechanisms complementary
  • Lines 67-71 – it would be helpful to clearly state the hypotheses here, in addition to describing the study aims
  • Line 77 – how was sample size determined? Was an a priori power analysis conducted?  If so, how was effect size and variance estimated?
  • Line 80 – did the subjects regularly use any recovery aid (e.g., foam roller, compression, massage etc.); if so, please quantify the usage and experience of subjects with these recovery aids
  • Line 86 – some measure of body composition should be reported; given that both foam roller and vibration roller effects are likely influence by body fat, skin thickness, subcutaneous fat deposits, and other factors related to body composition, this is a potential confounder or covariate that should be adjusted for; even simple skinfold measurements would be useful
  • Line 87 – how was randomization conducted? Were groups counterbalanced so the same number of subjects received FR or VR first?
  • Table 1. Please include more demographic data – body mass, weekly training volume, training history, body mass index, VO2max (if tested), etc.
  • Figure 1. The subject’s face should be blocked out to provide privacy, or a note should be made that the subject consented to having his photograph reported in this manuscript
  • Line 97 – was testing conducted at the same time of day for each visit?
  • Line 101 – is this supposed to say each leg, rather than each foot?
  • Lines 107-108 – why was the mixed frequency mode selected for the study, as opposed to other modes?
  • Line 112 – how was the laser Doppler flowmeter calibrated? Was the residual biological zero signal corrected for?  Rather than reporting in arbitrary laser Doppler units, why were values not normalized to values measured during maximal vasodilation (local skin warming or local sodium nitroprusside), and reported in mL/min?
  • Line 115 – how was the midpoint of the gastrocnemius determined and reproduced? Which anatomical landmarks were used, and how the site was marked and reproduced should be described
  • Line 116 – this should read leg, rather than foot
  • Line 116 – the position of the subject should be described, and how motion of the subject was minimized and corrected for since motion artifact is a major technical limitation in laser Doppler flowmetry
  • Line 137 – the use of multiple t-tests appears to be inappropriate here given that no correction was made for multiple comparisons. This likely increases the type I error rate to an unacceptably high level without proper adjustment
  • Table 2 – Foam roller is misspelled “form roller”
  • Table 4 – Foam roller is misspelled “form roller”
  • Table 5 – Foam roller is misspelled “form roller”
  • Table 6 – Foam roller is misspelled “form roller”
  • Table 5 – It is unclear why in Table 4, differences are expressed in absolute difference units, but then in Table 5, the pair differences are given as % change. Please justify this decision.  If pair differences are analyzed in absolute values, rather than % change, does that influence the results?
  • Line 162 – I’m not sure why the “significant improvement” in skin blood flow perfusion from pre-run to post-run in the FR group is relevant, because at the post-run time point, the participants had not yet received the intervention yet, so this “change” is not reflective of any treatment effect
  • Line 183-184 – the text states that there was a significant effect in normalized myogenic energy of relative change between post-run and post-roller for the VR group, but this data doesn’t appear to be presented – Table 8 appears to show a significant difference between FR and VR in pre-run vs post-roller pair difference, but I do not see the relative change between post-run and post-roller for the VR group with statistics (the raw data appears to be in Table 6, but no statistics are given)
  • Line 187-188 – while it does appear there is a numerical decrease in skin blood flow post-roller compared to post-run, Table 2 shows the P-value is not statistically significant, so I question why the authors are interpreting this as a significant finding
  • Line 195 – I do not believe this study found a statistically significant benefit from VR
  • Lines 205-216 – I am not sure why this paragraph is relevant; suggest revising to connect to the present study, or removing
  • Lines 230-235 – this limitations discussion should be expanded, and perhaps put as its own sub-heading
  • What is the physiologic and applied significance of a change in skin blood flow perfusion?
  • Lines 238-239 – This conclusions is unclear given that Table 4 shows no statistically significant differences in VR blood flow perfusion at any time point; Table 5 appears to show a difference between FR and VR in pre-run vs post-roller pair difference; please clarify exactly what the difference was, and revise accordingly
  • Line 239 – please remove “endothelial” from the conclusion – none of the data presented show a statistically significant difference in any of the endothelial measures
  • Line 240-241 – I don’t see evidence supporting “eliminate metabolic waste after exercise” – the data presented are simply on blood flow, and no blood samples were taken to analyze for blood concentrations of metabolites

Reviewer 4 Report

The main aim of the present study was to determine the different effects of foam rollers and vibrating rollers on skin blood flow after a running fatiguing task

THe research question is interesting however I have some major comments that I hope can help improve the quality of the manuscript

My major concern is that there is no task (or performance measurement) post roller application. So, independently of SBF measurement it would have been interesting to evaluate performance as well to assess recovery

Secondly, regarding the methodology of foam roller or vibrating roller, were subjects followed by a personal trainer? How is it possible to normalize these 2 recovey methods? Would changing time of application of one or the other change the recovery effects and SBF? I think this is a bit of a flaw in the methodology

Moreover, I would suggest to change tables in graphs. Easier for a reader to viausalize the main results imemdiateley. 

Thereafter, I would suggest to modify your discussion according to the points above described. IN the discussion, the first paragraph is a bit confusing. It seems that from previous studies, FR involves insufficient compression while later on you state that VR produces less pain. 

The conclusions are not supported by the data because you did not really measure fatigue (and/or performance)

Round 2

Reviewer 1 Report

I have much appreciated authors' work in modifying and improving the article and so, I think it is now suitable for conidering publication by editorial team.

Reviewer 4 Report

The manuscript has greatly improved, although I would have preferred a measure of fatigue and performance